# GROUP LIGANDS DOCKING TO PROTEIN POCKETS

**Jiaqi Guan**[1]*, **Jiahan Li**[2]*, **Xiangxin Zhou**[3], **Xingang Peng**[4],
**Sheng Wang**[5], **Yunan Luo**[6], **Jian Peng**[7], **Jianzhu Ma**[2]
[1] University of Illinois Urbana-Champaign, [2] Tsinghua University,
[3] University of Chinese Academy of Sciences, [4] Peking University,
[5] University of Washington, [6] Georgia Institute of Technology, [7] Helixon Research
jiaqi@illinois.edu, majianzhu@tsinghua.edu.cn

## ABSTRACT

Molecular docking is a key task in computational biology that has attracted increasing interest from the machine learning community. While existing methods have achieved success, they generally treat each protein-ligand pair in isolation. Inspired by the biochemical observation that ligands binding to the same target protein tend to adopt similar poses, we propose GROUPBIND, a novel molecular docking framework that simultaneously considers multiple ligands docking to a protein. This is achieved by introducing an interaction layer for the group of ligands and a triangle attention module for embedding protein-ligand and group-ligand pairs. By integrating our approach with diffusion-based docking model, we set a new S performance on the PDBBind blind docking benchmark, demonstrating the effectiveness of our proposed molecular docking paradigm.

## 1 INTRODUCTION

The 3D binding structures of molecular ligands and protein pockets are not only fundamental to unraveling the intricate protein-ligand interactions but also play a pivotal role in rational drug design. In this context, the molecular docking task, which involves predicting the optimal binding configuration between a ligand and a protein, stands as a cornerstone in modern computational biology. Conventional computational approaches leveraging the pre-defined score functions to search or optimize a low-energy pose configuration (Halgren et al., 2004; Trott & Olson, 2010) are usually slow and inaccurate, which limits their applicability in real-world drug discovery scenarios. In recent years, deep learning-based algorithms (Stärk et al., 2022; Lu et al., 2022; Zhang et al., 2022; Corso et al., 2022) have showcased remarkable success in accurately predicting the protein-ligand binding structures, which brings new breakthroughs in the field of molecular docking.

While these deep learning methods vary in modeling architectures and objectives, they all treat each protein-ligand pair *individually*. Given the limited number of well-studied protein targets compared to the vast space of molecules, many ligands share common protein targets in available databases. Exploiting the correlated information among these complexes could enhance docking performance.

In this work, we leverage a key biochemical insight: if multiple ligands can bind to the same protein pocket, their docked 3D structures are likely to be similar (Paggi et al., 2021). For instance, Figure 1 shows four ligands binding to the same target protein (UniProt ID: B1MDI3), all exhibiting very similar docking poses. This similarity arises from the conserved interactions within the binding site, suggesting that the protein pocket maintains a consistent set of key interactions that facilitate binding. For example, ligands often form similar hydrogen bonds with specific protein atoms, resulting in low-energy complexes and analogous spatial arrangements when docked to the same protein.

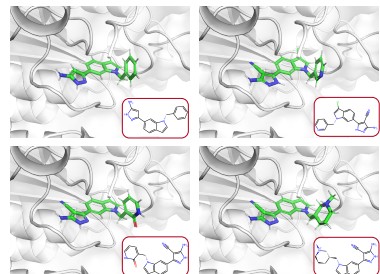

Figure 1: Multiple ligands that can bind to the same protein pocket exhibit similar docking poses.

---

*Equal Contribution

To harness the information shared among ligands binding to the same protein pocket, we propose GROUPBIND, a novel molecular docking paradigm that considers multiple ligands docking simultaneously. When docking a ligand $A$ to a protein pocket $P$, we can utilize other ligands in the same binding group to enhance docking performance of $A$ to $P$, as they may share similar poses and key interaction patterns. In other words, we can dock a group of ligands to the same protein pockets at the same time. Specifically, we develop an interaction layer that enables message passing among group ligands and a triangle attention module for pair representations, based on interaction consistency between different ligands and the protein pocket. We also encode specific structural information into these representations by providing additional supervised signals during training. By integrating our approach into a diffusion-based docking model (Corso et al., 2022), we achieve new state-of-the-art docking performance on the PDBBind benchmark.

To summarize, we highlight our contributions as follows: (1) We propose a novel paradigm for molecular docking by considering multiple ligands docking to the protein pocket based on a physical intuition. (2) We develop a new model, comprising modules which capture protein-ligand interaction consistency. This is the first time the concept that multiple molecules should yield similar docking poses has been introduced into an end-to-end deep neural network approach to the molecular docking task. (3) We provide a specific algorithmic implementation of the new paradigm mentioned above, for the first time verifying the feasibility of the idea that using multiple similar molecules to dock into a protein pocket simultaneously can enhance the docking accuracy.

## 2 RELATED WORK

**Molecular Docking.** Molecular docking aims at predicting the binding structure of a small molecule ligand to a protein, which is a critical technique in drug discovery and enables understanding of the intermolecular interactions and potential binding modes (Ferreira et al., 2015). Traditional molecular docking approaches usually design scoring functions and then adopt search, sampling, or optimization algorithms (Shoichet et al., 1992; Meng et al., 1992; Trott & Olson, 2010). The integration of deep learning techniques has seen advancements in enhancing scoring functions (Ragoza et al., 2017; McNutt et al., 2021; Méndez-Lucio et al., 2021). Recently, with the development of geometric deep learning, direct prediction of the binding pose has become an emerging paradigm and achieved promising results. As representative examples, TANKBind (Lu et al., 2022) and Uni-Mol (Zhou et al., 2023) predict protein-ligand distance map and using post-optimization algorithms to recover the pose from the predicted distance map, while EquiBind (Stärk et al., 2022) and E3Bind (Zhang et al., 2023) directly predict the coordinates of the ligand binding pose. FABind (Pei et al., 2024) and FAbind+(Gao et al., 2024) enhance docking accuracy by directly introducing a pocket prediction module. Among these, Corso et al. (2022) framed molecular docking as a generative modeling problem and proposed DiffDock, a diffusion model (Ho et al., 2020; Song et al., 2021) that learns the distribution over the product space of the ligand's translation, rotation, and torsion, along with a confidence model to pick up the final prediction from sampled ligand poses. Although recent physics-based docking methods (Paggi et al., 2021; McNutt & Koes, 2024) have utilized similar binding molecules or poses to enhance docking accuracy, our proposed GroupBind is the first deep learning-based method capable of integrating with existing docking frameworks and improving performance by leveraging group binding data.

**Consistency in Data.** Consistency is a common nature of biological data. This essential feature provides us insights to better understand function, structure, and evolution in biology. Conserved sequences, maintained by natural selection, are identical or similar sequences in nucleic acids or proteins across species, and many multiple sequence alignment (MSA) algorithms, such as ProbCons (Do et al., 2005) and CONTRAlign (Do et al., 2006), are developed to reveal such patterns. Liao et al. (2009); Hashemifar et al. (2016) developed effective protein-protein interaction (PPI) network alignment algorithms, which help identify conserved subnetwork and thus enable accurate identification of functional orthologs across species. Consistency also exists in other domains beyond biology and such methodology of mining consistency from data has also been extensively studied, e.g., network alignment (i.e., finding the node correspondence across different networks) which is of great value in social networks analysis and web mining (Zhang & Tong, 2016; Zhang et al., 2021).

**Consistency-Inspired Models.** By virtue of its ubiquity and usefulness, consistency serves as a prior knowledge that plays crucial roles in various areas. Xiao & Quan (2009) and Hu et al. (2020)

leveraged the visual consistency among different views of the same object to improve the semantic segmentation and 3D shape completion, respectively. Zheng et al. (2021) proposed consistency regularization for cross-lingual fine-tuning inspired by the belief that the answers of semantically similar questions are supposed to be consistent. Zhou et al. (2003) introduced the assumption prior that samples in the same cluster tend to have the same labels to semi-supervised learning. Consistency also exists in the protein-ligand docking task: different ligands tend to form similar interactions with the protein. Inspired by this intuition, we propose a new module capturing such protein-ligand interaction consistency.

## 3 METHOD

In this section, we present GROUPBIND, a new and effective paradigm that leverages other ligands binding to the same protein pocket to improve single ligand docking accuracy, where we dock group ligands to this pocket at the same time. In Section 3.1, we first define notations and briefly recap the diffusion model for molecular docking, which we will integrate our paradigm into as an example in this work. Then, we introduce the modeling objective of GROUPBIND in Section 3.2. In Section 3.3, we illustrate how we parameterize GROUPBIND with message passing across group ligands and triangle attention between group ligands and proteins. Finally, in Section 3.4, we describe the auxiliary distance prediction during training and the confidence model for group ligands.

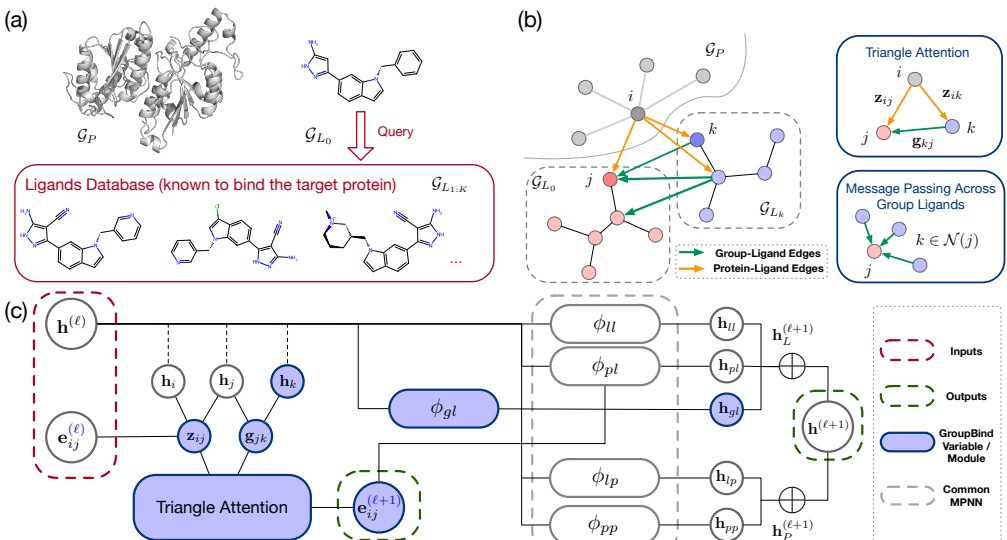

Figure 2: Overview of the proposed GroupBind paradigm. (a) GroupBind leverages other ligands $\mathcal{G}_{L_{1:K}}$ known to bind the same target protein $\mathcal{G}_P$ to facilitate the docking of $\mathcal{G}_{L_0}$. (b) GroupBind introduces the message passing across group ligands and the triangle attention module between group ligands and proteins. (c) GroupBind is not limited to any specific docking framework and can be integrated into existing models (dashed gray box) with the aforementioned modules (blue ones).

### 3.1 PRELIMINARIES

**Problem Definition and Notation.** In the molecular docking task, we are provided with a protein (pocket) $P$ with $N_P$ residues and a ligand $L$ with $n$ atoms. Our goal is to find the docking pose $\mathbf{x} \in \mathbb{R}^{3n}$ of the ligand in this binding site. Following previous work (Lu et al., 2022; Corso et al., 2022), we represent a protein pocket as a residue-level proximity 3D graph $\mathcal{G}_P = \{\mathcal{V}_P, \mathcal{E}_P\}$, where $\mathcal{V}_P$ denotes the residue vertex set consisting of residue vertex features $\mathbf{v}_P \in \mathbb{R}^{N_P \times N_a}$ (such as amino acid types) and $C_\alpha$ atom positions of residues $\mathbf{x}_P \in \mathbb{R}^{N_P \times 3}$, and $\mathcal{E}_P$ denotes the edge set which is built upon residue 3D proximity. Similarly, the small molecule ligand is also represented as a 3D graph $\mathcal{G}_L = \{\mathcal{V}_L, \mathcal{E}_L\}$. Since bond lengths, angles, and small rings in the ligand can be regarded as rigid structures, the torsion angles almost determine the flexibility of ligand conformations $\mathbf{x}$. Thus, we can also define the space of ligand poses on a submanifold $\mathcal{M}_\mathbf{c} \in \mathbb{R}^{m+6}$ instead of $\mathbb{R}^{3n}$ given the seed conformation $\mathbf{c}$ (Corso et al., 2022), where $m$ is the number of rotatable bonds in the

ligand and the six additional degrees of freedom correspond to the roto-translation relative to the protein pocket.

**Diffusion Model for Molecular Docking.** Recently, DiffDock (Corso et al., 2022) proposed to formulate molecular docking as a generative task, with the objective of modeling the conditional distribution $p(g|\mathcal{G}_P, \mathcal{G}_L)$ and $g \in \mathcal{M}_{\mathbf{c}}$. For a given seed conformation $\mathbf{c}$, we can define a bijection $\mathbb{P} \to \mathcal{M}_{\mathbf{c}}$, where $\mathbb{P} = \mathbb{T}(3) \times SO(3) \times SO(2)^m$ is the production space of ligand translation $\mathbf{r}$, rotation $R$, and torsion angles $\theta$. Leveraging recent progress of diffusion models on Riemannian manifolds (De Bortoli et al., 2022), we can define the forward diffusion independently in each manifold, resulting in a direct-sum tangent space: $T_g\mathbb{P} = T_{\mathbf{r}}\mathbb{T}_3 \oplus T_R SO(3) \oplus T_\theta SO(2)^m \cong \mathbb{R}^3 \oplus \mathbb{R}^3 \oplus \mathbb{R}^m$. Based on this formulation, we can train the diffusion model by score matching on translational scores, rotational scores and torsional scores, and sampling ligand poses by running the reverse diffusion process (Song et al., 2021).

## 3.2 MODELING GROUP LIGANDS

Existing deep learning-based models for molecular docking consider each protein-ligand pair individually. However, one important biological observation suggests that when multiple ligands are capable of binding to a common protein pocket, the resulting docking poses are expected to exhibit a significant degree of similarity. To include this inductive bias, we propose to model the docking poses of a *group* of ligands given the common protein pocket, instead of modeling them individually. Specifically, we are docking a ligand $\mathcal{G}_{L_0}$ along with $k$ other ligands of interest $\mathcal{G}_{L_1}, \ldots, \mathcal{G}_{L_K}$ to a shared protein pocket $\mathcal{G}_P$. The $K$ other ligands $\mathcal{G}_{L_{1:K}}$ could be the ones whose docking poses we are also interested in, or retrieved from some database as the *augmented ligands* to help us dock $\mathcal{G}_{L_0}$. Importantly, we do *not* utilize the binding structural information of augmented ligands $\mathcal{G}_{L_{1:K}}$ during the inference phase, but only leverage these ligands known to bind to the same protein to improve single protein-ligand docking performance. Under this formulation, our modeling objective becomes the product space $\mathbb{P}_g$ of translation, rotation and torsion angles of this group of ligands, i.e.

$$p\left((\mathbf{r}_{0:K}, R_{0:K}, \theta_{0:K})|\mathcal{G}_P, \mathcal{G}_{L_{0:K}}\right), \tag{1}$$

where $\mathbb{P}_g = \mathbb{T}(3)^{(K+1)} \times SO(3)^{(K+1)} \times SO(2)^{\sum_{k=0}^{K} m_k}$.

Considering the effectiveness of generative models in molecular docking, such as diffusion models (Corso et al., 2022), we choose to adopt a diffusion framework to model the binding distribution. However, our modules can be integrated into other frameworks that use pairwise representations of molecules. The key technical challenge lies in enabling each ligand to become aware of other ligands in the group to improve docking performance. In the following, we will explain how we address this by constructing graphs among group ligands and introducing novel network architectures that allow the model to account for interactions within the group.

## 3.3 PARAMETERIZATION OF GROUPBIND

**Construct Ligand Group.** Ligands are grouped together when they share the same protein binding pocket, as outlined in Algorithm 1 in Appendix. We group complexes based on the protein's UniProt ID and name provided in the original PDBBind dataset. To align proteins within the same group, we use the Kabsch algorithm (Kabsch, 1976) to compute the optimal roto-translation based on the longest common subsequence of amino acids. The reference protein is chosen as the one with the lowest average root-mean-square deviation (RMSD) from others in the group. The resulting optimal rigid transformations are applied to both the protein and ligand in other complexes. To ensure high-quality training data, we also filter out complexes with protein-ligand minimum distances that are either too close or too far.

**Build Group Ligands Graph.** To facilitate the exchange and aggregation of useful docking information, we construct a graph across ligands in the group by placing all group ligands in the same coordinate system and adding edges between different ligands. This design facilitates the exchange and aggregation of useful docking information among ligands.. A straightforward approach to constructing the group ligands graph is to dynamically build a radius graph or $k$-NN graph based on the diffused group ligands, which are the ligands with added noise during the diffusion process. This is similar to how the protein-ligand heterogeneous graph is constructed. However, in scenarios where the noise level is high, the edges formed based on 3D proximity relative to the protein may

inadvertently link two atoms in different ligands with entirely distinct interactions with the protein pocket. In such cases, the message passing process among group ligands might not yield helpful insights and, in fact, could have adverse effects. Conversely, considering the computational resource constraints, creating a fully-connected graph for group ligands and discerning which connections are pertinent from the data is also impractical. [1]

Given that ligands binding to the same protein pocket typically exhibit some degree of structural similarity, we propose an approach to construct the group ligands graph that is *independent* of the protein pocket. This independence implies that the construction of the group ligands graph is solely based on the structural and 3D geometry similarities among ligands. To achieve this, we initially conduct 2D molecular graph matching, establishing a one-to-one mapping through the Maximum Common Substructure (MCS) (Raymond & Willett, 2002). For those unmatched atoms, we build edges by connecting their neighbors within 4Å if these ligands are presented as training data. Otherwise, i.e. in the inference phase, we perform 3D matching (Wildman & Crippen, 1999) on generated molecular conformers and similarly connect neighbors within a 4Å radius. The edges among group ligands are independent of the diffusion time, and since the C-C bond length is about 1.5Å. The 4Å is chosen by assuming each atom can interact with corresponding 2-hop neighbors.

**Message Passing across Group Ligands.** To make group ligands aware of each other, we introduce a message passing mechanism across group ligands based on the aforementioned group ligands graph. Similar to DiffDock, we utilize irreducible representations (irreps) of $SO(3)$ (Thomas et al., 2018; Fuchs et al., 2020) to construct the message passing layer, implemented with the e3nn library (Geiger & Smidt, 2022). Denoting the group ligands graph as $\mathcal{A}$, the node features as $\mathbf{h}_j$ and $\mathbf{h}_k$, where $j, k$ represent atoms from two different ligands, with $k$ being in the neighborhood of $j$ within the group ligands graph, i.e. $\{j \in \mathcal{V}_{L_g}, k \in \mathcal{V}_{L_{g'}} \mid g \neq g', (j,k) \in \mathcal{E}_{\mathcal{A}}\}$, the message passing layer for each ligand atom $j$ within group ligands at $\ell$-th layer is then defined as follows:

$$\Delta_{\mathcal{A}} \mathbf{h}_j^{(\ell)} = \text{BN}\left( \frac{1}{|\mathcal{N}_{\mathcal{A}}(j)|} \sum_{k \in \mathcal{N}_{\mathcal{A}}(j)} Y(\hat{r}_{jk}) \otimes_{\psi_{jk}} \mathbf{h}_k^{(\ell)} \right), \tag{2}$$

$$\psi_{jk} = \phi_e([\mathbf{h}_j^{(\ell)}, \mathbf{h}_k^{(\ell)}, \mathbf{e}_{jk}]),$$

where BN indicates the batch normalization layer, $Y(r_{jk})$ are the spherical harmonics of edge vector $r_{jk}$ up to $l = 2$, and $\otimes_{\psi_{jk}}$ denotes the spherical tensor product with path weights $\psi_{jk}$, which is obtained through a MLP $\phi_e$ taking as input the edge embedding $\mathbf{e}_{jk}$ and *scalar* node features. The node feature update $\Delta_{\mathcal{A}} \mathbf{h}_j^{(\ell)}$ will be combined with other updates derived from the base docking neural network to obtain the updated the ligand node feature $\mathbf{h}_L^{(\ell+1)}$, as shown in Figure 2.

**Triangle Attention Between Group Ligands and Protein.** While the above message passing mechanism establishes connections between different ligands within the group, it lacks consideration for the interaction consistency with the protein. Inspired by AlphaFold2 (Jumper et al., 2021), we introduce a triangle attention module upon pair embeddings to incorporate geometric consistency. It is worth noting that Lu et al. (2022); Zhang et al. (2022) also employ triangle attention modules to enforce geometric constraints. However, a key distinction lies in our attention module operates *across different ligands*, reflecting a fundamentally different underlying motivation: if atom $j$ in ligand $L_g$ can interact with amino acid $i$, then the corresponding atom $k$ in a similar ligand $L_{g'}$ should also interact with amino acid $i$.

To incorporate this idea, we first construct pair representations $\mathbf{z}_{ij}$ for protein-ligand edges, and $\mathbf{g}_{jk}$ for group ligand-ligand edges, which are both obtained from MLPs with scalar node features and edge embeddings:

$$\mathbf{z}_{ij}^{(\ell)} = \phi_c([\mathbf{h}_i^{(\ell)}, \mathbf{h}_j^{(\ell)}, \mathbf{e}_{ij}^{(\ell)}]),$$
$$\mathbf{g}_{jk}^{(\ell)} = \phi_g([\mathbf{h}_i^{(\ell)}, \mathbf{h}_j^{(\ell)}, \mathbf{e}_{jk}]). \tag{3}$$

When ligand atoms $j$ and $k$ exhibit structural similarity, it is expected that they would engage in similar interactions with any protein atom $i$. To capture this, we implement a triangle multi-head

---

[1]As we will see below, the computational cost of triangle attention is $\mathcal{O}(|\mathcal{E}_{pl}| \times |\mathcal{E}_{gl}|^2)$, where $|\mathcal{E}_{pl}|$ is the number of edges between protein and ligands, and $|\mathcal{E}_{gl}|$ is the number of edges between group ligands. A fully-connected graph will induce a large number of $|\mathcal{E}_{gl}|$ and thus a high computational cost.

attention mechanism involving atoms $i, j, k$ to update pair features $\mathbf{z}_{ij}$. The attention weight is controlled by $\mathbf{g}_{jk}$, leading to the following formulation in Equation 4:

$$\mathbf{z}_{ij}^{(\ell)} = \mathbf{z}_{ij}^{(\ell)} + \text{Linear}\Big(\text{concat}_h\Big(\sum_k s_{ijk}^{(\ell)h} \mathbf{v}_{ik}^{(\ell)h} \phi(\mathbf{z}_{ij}^{(\ell)h})\Big)\Big),$$

$$s_{ijk}^{(\ell)h} = \text{softmax}\Big(\frac{1}{\sqrt{c}}\mathbf{q}_{ij}^{(\ell)h\top}\mathbf{k}_{ik}^{(\ell)h} + b_{kj}^{(\ell)h}\Big), \tag{4}$$

where $\mathbf{q}_{ij}, \mathbf{k}_{ik}, \mathbf{v}_{ik}$ are linear transformations of protein-ligand pair embedding $\mathbf{z}$, and $b_{kj}$ is the linear transformation of group ligand pair embedding $\mathbf{g}_{kj}$. $\phi$ is a gated linear transformation and $c$ is the dimensionality of the query / key / value features.

Similarly, we can exchange the roles of key embedding $\mathbf{k}_{ik}$ and attention bias $b_{kj}$, resulting in a new attention module with group pair embedding as attention keys, which is stacked with the aforementioned attention module to improve the model capacity:

$$\mathbf{z}_{ij}^{(\ell)} = \mathbf{z}_{ij}^{(\ell)} + \text{Linear}\Big(\text{concat}_h\Big(\sum_k s_{ijk}^{(\ell)h} \mathbf{v}_{kj}^{(\ell)h} \phi(\mathbf{z}_{ij}^{(\ell)h})\Big)\Big),$$

$$s_{ijk}^{(\ell)h} = \text{softmax}\Big(\frac{1}{\sqrt{c}}\mathbf{q}_{ij}^{(\ell)h\top}\mathbf{k}_{kj}^{(\ell)h} + b_{ik}^{(\ell)h}\Big). \tag{5}$$

Finally, we update the pairwise protein-ligand edge features with the pair representation $\mathbf{z}_{ij}^{(\ell)}$ and the initial edge embedding $\mathbf{e}_{ij}^0$ through MLPs:

$$\mathbf{e}_{ij}^{(\ell+1)} = \phi_2([\mathbf{e}_{ij}^{(0)}, \phi_1(\mathbf{z}_{ij}^{(\ell)})]). \tag{6}$$

In this way, we introduce a pairwise protein-ligand edge hidden variable $\mathbf{e}_{ij}^{(\ell)}(l = 0, \ldots, L - 1)$ which is updated through layers, *instead of* regarding them as fixed edge embeddings. This allows effective message passing between protein and group ligands through layers.

## 3.4 Training and Inference

**Auxiliary Distance Prediction.** By constructing $\mathbf{e}_{ij}$ as a hidden variable, we can encode more structural information into it, not merely using it as an edge feature. We propose to predict the binned protein-ligand distances from the pair representations during the training phase, i.e. $\hat{d}_{ij}^b = \phi_d(\mathbf{e}_{ij}^L)$. It provides the supervised signal to make pairwise features encode specific geometric information. Since in this work, we integrate GroupBind with the DiffDock framework, the auxiliary training loss becomes a weighted cross entropy loss based on the diffusion time $t$:

$$\mathcal{L}_{\text{dist}} = \frac{1}{N_{\text{pairs}}} e^{-t} \sum_{i,j} \sum_{b=1}^{B} d_{ij}^b \log \hat{d}_{ij}^b, \tag{7}$$

where $B$ is the total number of distance bins.

**Confidence Model.** Following DiffDock, we also train a confidence model to perform ranking on sampled ligand poses. We run the trained diffusion model to collect a set of candidate poses for each target protein and generate the binary label by checking whether the ligand RMSD is less than 2 Å. Notably, unlike DiffDock, our confidence model involves ranking a *group* of ligand poses that can bind to the same protein. In the preliminary experiments, we observed that training the confidence model to directly predict the group score did not yield optimal results. Instead, training the confidence model to predict individual scores and subsequently computing the group score by averaging them proved more effective, i.e. the group confidence is

$$\mathbf{c}(\mathbf{x}_0, \ldots, \mathbf{x}_K) = \frac{1}{K} \sum_{k=0}^{K} \phi_c(\mathbf{x}_k),$$

where $\phi_c$ is the confidence model, and we rank ligand poses $(\mathbf{x}_0, \ldots, \mathbf{x}_K)$ as a group by the group confidence $\mathbf{c}(\mathbf{x}_0, \ldots, \mathbf{x}_K)$ instead of ranking them individually.

**Network Implementation.** Following TANKBind Lu et al. (2022), we also apply the ML-based ligand binding sites prediction algorithm P2Rank Krivák & Hoksza (2018) to segment the protein into potential binding pockets, which is mainly due to computational memory considerations. Additionally, in most real application scenarios, the ligand binding site is known. We set the pocket radius as 20 Å, which is large enough to enclose the ligand molecules in most cases. The denoising network consists of embedding layers, interaction layers and output layers following DiffDock, and each interaction layer is combined with our proposed group-ligand interaction layer and triangle attention layer between group ligands and protein. Since our proposed modules introduce additional parameters, we use five interaction layers instead of six, resulting in 18.8 million trainable parameters, which is on par with 20.3 million parameters of DiffDock.

**Training Consideration.** During the training phase, we set the maximum number of ligands in the group (in which the ligands can bind to the same target protein) as 5. If there are more than five ligands binding to the same protein, we perform the agglomerative clustering Defays (1977) on ligands based on their Tanimoto similarity Bajusz et al. (2015) to make sure each group has less than or equal to 5 ligands. We select ligands whose center of mass is less than 8 Å from the center of native pocket or P2Rank predict pockets as training data.

**Inference Procedure.** During inference, we do not rely on 3D structural data of the ligands. Instead, we leverage 2D representations of other ligands known to bind to the same protein to improve the docking performance of the query ligand. Given a protein and a query ligand, we search the database (in our case, the PDBBind dataset) to identify ligands that bind to the same protein. If such ligands are found, they are incorporated into the docking process alongside the query ligand. The resulting docking pose of the query ligand is then used as the final output. If no such ligands are available, our approach defaults to the standard single-ligand docking settings, as explored in prior studies.

## 4 EXPERIMENTS

### 4.1 EXPERIMENTAL SETUP

**Dataset.** We evaluate our method on PDBBind v2020 dataset (Liu et al., 2015), which contains structures of 19443 protein-ligand complexes collected from PDB (Berman et al., 2003). We use the same processed proteins as Stärk et al. (2022), which uses `reduce`[2] to correct and add missing receptor hydrogens. We use the same time-based split following previous work (Stärk et al., 2022; Lu et al., 2022; Corso et al., 2022), resulting in 17k complexes from 2018 or earlier for training/validation and 363 complexes from 2019 with no ligand overlap for testing.

**Baselines.** We compare our model with search-based methods SMINA Koes et al. (2013), GNINA McNutt et al. (2021), and various state-of-the-art deep learning methods EquiBind Stärk et al. (2022), TANKBind Lu et al. (2022) and DiffDock Corso et al. (2022). For searching-based methods SMINA and GNINA, we also test their variants by combining them with P2Rank or EquiBind to identify the initial binding site first.

**GroupBind.** We test the performance of GroupBind under different settings, denoted as different postfixes: 1. **Ref**: Utilizes the native protein pocket, where the pocket center is defined as the centroid of the centers of mass (CoMs) of the group ligands. 2. **P2R**: Uses the top-3 P2Rank predicted pockets. For each pocket, we sample the same number of ligand poses and apply our confidence model to select the top-$K$ ligands across all poses. 3. **S(G)**: Groups ligands within the test set for self-augmentation, denoted as self-augment variant ("-S"). The 363 test ligands form 86 groups (255 ligands) with more than one ligand in each group, and 108 groups with single ligand. 4. **A(G)**: Expands the augmented ligand database using ligands from the training set, allowing for the augmentation of 32 out of the 108 single ligands. 5. **N(G)**: No augmented ligand database is used, only the original ligands are utilized for docking.

### 4.2 IMPROVING DOCKING PERFORMANCE WITH GROUPBIND

Following prior work Stärk et al. (2022); Lu et al. (2022); Corso et al. (2022), in assessing the quality of the generated complexes, we calculate the permutation symmetry corrected Meli & Biggin (2020)

---

[2]`https://github.com/rlabduke/reduce`

Table 1: Summary of top-1 ligand RMSD and top-5 ligand RMSD metrics of baselines and GROUP-BIND variants. Baseline results are from the original DIFFDOCK paper, except the DIFFDOCK itself, which we perform sampling following the official repository and report the evaluation results. (↑) / (↓) denotes a larger / smaller number is better. The best results are highlighted with **bold text** (The GROUPBIND-REF variants are not taken into consideration for a fair comparison).

| | Top-1 Ligand RMSD | | | | | Top-5 Ligand RMSD | | | | |
| | Percentiles (↓) | | | % < thres. (↑) | | Percentiles (↓) | | | % < thres. (↑) | |
| Method | 25th | 50th | 75th | 5Å | 2Å | 25th | 50th | 75th | 5Å | 2Å |
|---|---|---|---|---|---|---|---|---|---|---|
| GNINA | 2.4 | 7.7 | 17.9 | 40.8 | 22.9 | 1.6 | 4.5 | 11.8 | 52.8 | 29.3 |
| SMINA | 3.1 | 7.1 | 17.9 | 38.0 | 18.7 | 1.7 | 4.6 | 9.7 | 53.1 | 29.3 |
| TANKBIND | 2.5 | 4.0 | 8.5 | 59.0 | 20.4 | 2.1 | 3.4 | 6.1 | 67.5 | 24.5 |
| P2RANK+SMINA | 2.9 | 6.9 | 16.0 | 43.0 | 20.4 | 1.5 | 4.4 | 14.1 | 54.8 | 33.2 |
| P2RANK+GNINA | 1.7 | 5.5 | 15.9 | 47.8 | 28.8 | 1.4 | 3.4 | 12.5 | 60.3 | 38.3 |
| EQUIBIND+SMINA | 2.4 | 6.5 | 11.2 | 43.6 | 23.2 | 1.3 | 3.4 | 8.1 | 60.6 | 38.6 |
| EQUIBIND+GNINA | 1.8 | 4.9 | 13.0 | 50.3 | 28.8 | 1.4 | 3.1 | 9.1 | 61.7 | 39.1 |
| DIFFDOCK | **1.6** | 3.6 | 7.9 | 59.3 | 32.4 | 1.4 | 2.4 | 4.8 | 75.6 | 41.3 |
| GROUPBIND-P2R-S | 1.7 | 3.4 | 7.9 | 57.9 | **33.2** | **1.3** | **2.1** | 4.2 | 77.8 | 47.4 |
| GROUPBIND-P2R-A | 1.8 | **3.2** | **6.6** | **66.2** | 32.1 | **1.3** | **2.1** | **4.0** | **80.1** | **48.8** |
| GROUPBIND-Ref-S | 1.6 | 2.8 | 5.8 | 69.3 | 36.6 | 1.2 | 2.0 | 3.9 | 82.3 | 50.7 |
| GROUPBIND-Ref-A | 1.6 | 2.9 | 5.9 | 70.4 | 36.3 | 1.2 | 2.0 | 3.6 | 84.5 | 49.0 |

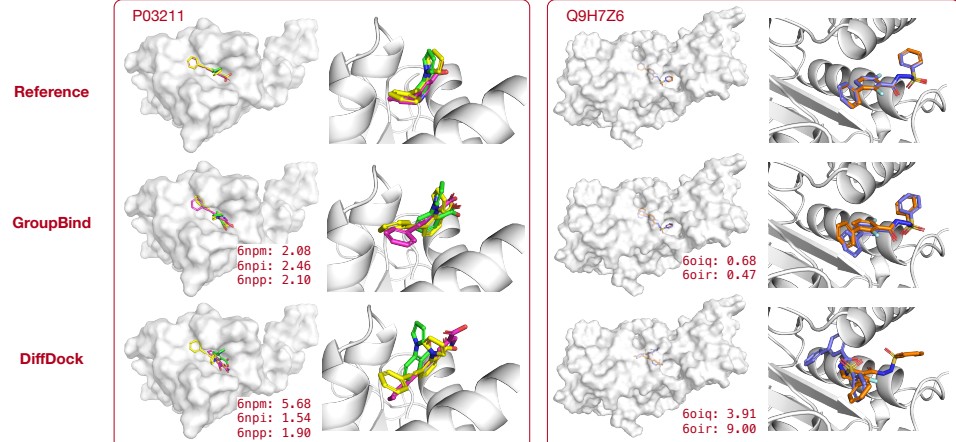

Figure 3: Visualization of two examples (UniProt ID: P03211, Q9H7Z6) with reference ligand docking poses, GroupBind's best predicted docking poses and DiffDock's best predicted docking poses. There are two and three different ligands that can bind to P03211 and Q9H7Z6, respectively, highlighted in distinct colors. Reference ligands tend to bind the target protein with similar poses, which is well captured with GroupBind and results in lower average RMSD compared to DiffDock.

heavy-atom RMSD between predicted and crystal ligand atoms. We report both 25th / 50th / 75th ligand RMSD percentiles and the percentages of predictions with an RMSD less than 2 Å or 5 Å.

All of baselines and GroupBind variants are able to generate multiple structures and rank them, and we sample 40 poses per pocket as the same setting of DiffDock. Results of the centroid distance can be found in the Appendix Section D. We also provide examples of docking poses generated by GroupBind compared to DiffDock in Figure 3.

We report the top-ranked prediction (Top-1) and the most accurate pose among the five highest ranked predictions (Top-5) in Table 1. It can be seen that GROUPBIND-P2RANK, the models in the full blind docking setting and thus fair to other baselines, can outperform all previous methods with a clear margin, especially in top-5 ligand RMSD. The self-augment variant ("-S") and training-augment variant ("-A") can achieve 47.4% and 48.8% with RMSD < 2 Å. We analyze the effects of using augmented ligand molecules comprehensively in Section 4.3. When provided with the native reference pocket, the top-1 ligand RMSD is significantly improved to 36.6% and 36.3% respectively. These results indicate that GroupBind can improve the docking performance by effectively leveraging the fact that multiple ligands tend to bind the same target protein in similar poses.

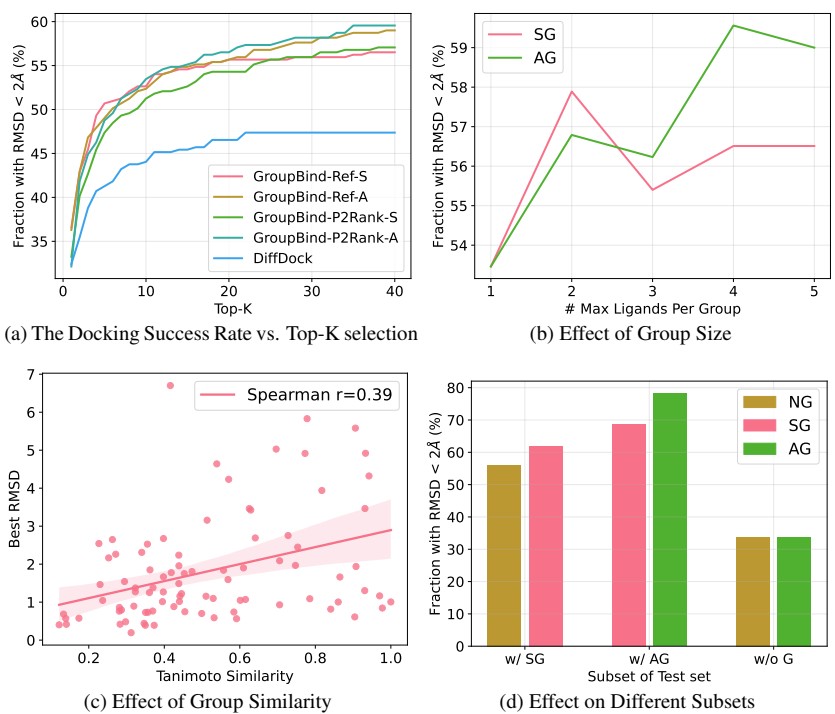

Figure 4: Analysis of the effect of augmented ligands. (a) The docking success rate (ligand RMSD < 2 Å) increases with the number of selection increases. (b) Effect of the group size. "SG" and "AG" indicate group ligands within test set and further utilize ligands from the training set respectively. (c) Effect of Group Similarity. X-axis denotes the Tanimoto similarity within the ligand group and Y-axis denotes the average best docking RMSD. (d) Effect on Different Subsets. "w/SG, w/AG and w/o G" indicates the subsets where augmented ligands from the test set, the training set, and no augmented ligands can be used respectively. "SG, AG, NG" indicates the corresponding model performance under different augmented ligands settings: test set, training set and no augmented ligands. For (b)(c)(d), the best docking pose among 40 candidates is applied for the success rate.

Taking a closer look at the top-$K$ fraction with ligand RMSD < 2 Å, we can see that our proposed GroupBind is consistently better than DiffDock for $K > 1$ and the margin becomes larger as $K$ increases, as shown in Figure 4a. With the perfect selection (equivalent to $K = 40$), GroupBind can achieve around 55% - 60% success rate while DiffDock can only achieve 47%. This indicates our model has a great potential to generate high-quality binding pose. Developing a more powerful confidence model is one promising direction to improve the docking performance further.

## 4.3 Effect of Augmented Ligands

We train GroupBind with the maximum number of ligands in each group as 5. However, we can assign a different maximum number of ligands during the sampling phase. In Figure 4b, we investigate the influence of using different numbers of augmented ligands in sampling. Similarly, we report the perfect selection results to isolate the influence of the confidence model. We can see that with augmented ligands, the success rate clearly improves, no matter the augmented ligands come from the test set itself (SG) or the whole dataset (AG). The optimal number of augmentation ligands does not strictly match the ones we used during training. Upon closer inspection of the structural similarity within each group, we find a subtle correlation between group similarity and the best ligand RMSD. This suggests that combining dissimilar ligands proves advantageous for our model in discerning correct interactions with proteins, ultimately yielding more accurate docking poses. Consequently, this highlights the pivotal role of the quality of augmentation ligands over their sheer quantity in enhancing the overall performance of our model. In Figure 4c, the presence of dissimilar ligands binding to the same pocket provides valuable information to our model, leading to improved docking accuracy and generally lower RMSD values (Spearman correlation = -0.39). This suggests that grouping dissimilar ligands is advantageous for our model in discerning correct interactions with proteins, ultimately yielding more accurate docking poses. In Figure 4d, we split the test set

into several subsets further to investigate where the performance improvement comes. The subsets are: (1) w/ SG, where ligands can be augmented with test set itself (255 PDBs), (2) w/ AG, where ligands can be further augmented by the training/validation set (32 PDBs) and (3) w/o G, where we can not find the augmented ligands in the whole dataset (76 PDBs). We can see the docking performance can be improved significantly if we can apply augmented ligands (w/ SG, w/ AG), while for the complexes where we cannot find other augmented ligands binding to the same protein, the performance remains unchanged and is typically hard to predict (only 30% perfect success rate), compared to other complexes (50% - 70% success rate).

## 4.4 ABLATION STUDIES

Table 2: Ablation studies on our proposed modules. "MP-GL" denotes the message passing across group ligands; "Att" denotes the triangle attention module; "Dist" denotes the auxiliary distance prediction. The results of perfect selection under 10 samples are reported.

| Method | MP-GL | Att | Dist | $\% < 2$Å | Median |
|---|---|---|---|---|---|
| DIFFDOCK | | | | 30.6 | 3.0 |
| | ✓ | | | 35.2 | 2.6 |
| GROUPBIND | ✓ | ✓ | | 38.8 | 2.5 |
| | ✓ | ✓ | ✓ | **42.9** | **2.2** |

To investigate the effectiveness of each component in GroupBind, we perform the ablation study as shown in Table 2. We employ a smaller neural network in both DiffDock and GroupBind variants, and the results of perfect selection under 10 samples are reported to isolate the influence of the confidence model. See more experimental setup details and full results in the Appendix Section C. It can be seen that each component can improve the docking performance substantially. One surprising thing is that even though the introduction of the auxiliary distance loss is straightforward, it can further improve the docking performance dramatically, proving the effectiveness of providing supervised signals to guide the learning of pairwise embedding.

## 4.5 DOCKING LIGANDS TO NEW PROTEINS

Table 3: Performance comparison between No Augmentation and Augmentation settings.

| | Top-1 Ligand RMSD | | | | | Top-5 Ligand RMSD | | | | |
|---|---|---|---|---|---|---|---|---|---|---|
| | 25th | 50th | 75th | $\% < 2$Å | $\% < 5$Å | 25th | 50th | 75th | $\% < 2$Å | $\% < 5$Å |
| No Aug | 2.3 | 4.9 | 24.9 | 20.5 | 50.0 | 1.5 | 3.2 | 6.8 | 34.1 | 68.2 |
| Aug | 1.8 | 4.7 | 15.5 | 29.6 | 50.0 | 1.5 | 3.4 | 6.3 | 38.6 | 65.9 |

As our method relies on ligands from the same binding group, we considered cases where no binding information is available for a new protein. In such cases, ligands binding to homologous proteins can be used as an alternative. To simulate this, we used FoldSeek (Van Kempen et al., 2024) (E-value $< 1e-5$) to identify homologous proteins and collected ligands (Tanimoto similarity $> 0.5$) binding to them for proteins without known ligands in PDBBind. These ligands were used as augmented inputs in our method. If neither ligands from the same protein nor homologous proteins are available, the method defaults to a diffusion-based docking model. Results in Table 3 demonstrate improved docking performance, showing our method's flexibility across various scenarios.

## 5 CONCLUSION

We introduce GROUPBIND for simultaneously docking multiple ligands to protein pockets, a novel paradigm for the molecular docking task grounded in sound physical intuition. We develop a novel model equipped with specialized modules to capture the consistency in protein-ligand interactions, holding the potential for seamless integration with diverse deep learning-based docking frameworks. Empirical results demonstrate the efficacy of our paradigm and the augmented group ligands. The limitation of our model lies in the high computation cost of the triangle attention module, rendering it impractical for constructing a relatively dense graph between group ligands and the protein. Furthermore, the proposed paradigm is contingent on the availability of group ligand binding information, presenting a limitation in scenarios where such data is absent. One potential future work is to leverage non-binding ligand information and construct a model that learns to selectively identify ligands beneficial for the docking process.

**Acknowledgement** This work was supported by the National Natural Science Foundation of China grants 62377030 and China's Village Science and Technology City Key Technology funding.

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

## A    CASE MOTIVATION

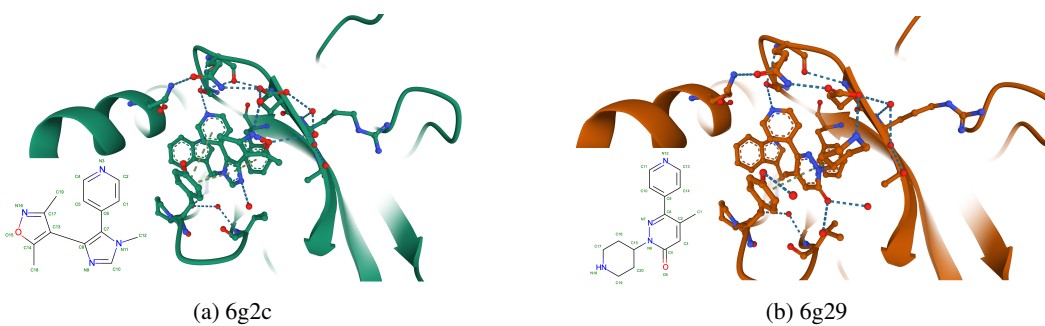

(a) 6g2c                                      (b) 6g29

Figure 5: PDB entries 6g2c and 6g29 contain distinct ligands (Tanimoto similarity = 0.26) that bind to the same pocket but share similar binding poses.

PDB entries 6g2c and 6g29 contain distinct ligands (Tanimoto similarity = 0.26) that bind to the same pocket. Despite their differences, several key interactions are conserved: for instance, a hydrogen bond is formed between N3 in 6g2c / N12 in 6g29 and an oxygen atom in the side chain of a pocket residue, while $\pi$-stacking interactions occur between the ring $C_7$-$C_8$-$N_9$-$C_{10}$-$N_{11}$ in 6g2c / ring $C_8$-$N_7$-$N_6$-$C_4$-$C_3$-$C_2$ in 6g29 and two benzene rings in the side chains of pocket residues. These conserved interactions ultimately result in similar binding poses for these distinct ligands.

## B    DATA PROCESSING DETAILS

We illustrate how we prepare the data for GROUPBIND in Section 1. After processing the original PDBBind v2020 dataset, we obtain 7323 groups (17649 complexes) for the training set, with 46.3 % groups having more than one ligand. By setting the maximum ligands in each group as 5, we have 8237 datapoints with 51.3 % datapoints with more than one ligand.

In Figure 6a, we present a histogram showing the number of ligands per pocket. There are 7697 unique pockets for the 19k complexes in the PDBBind dataset. Among these, 3,681 pockets (47.8%) contain more than one ligand. Additionally, Figure 6b illustrates the Tanimoto similarity of ligands within each group (with two or more ligands). This analysis shows that while some pockets are indeed bonded with similar ligands, a larger portion of pockets contain ligands that are dissimilar.

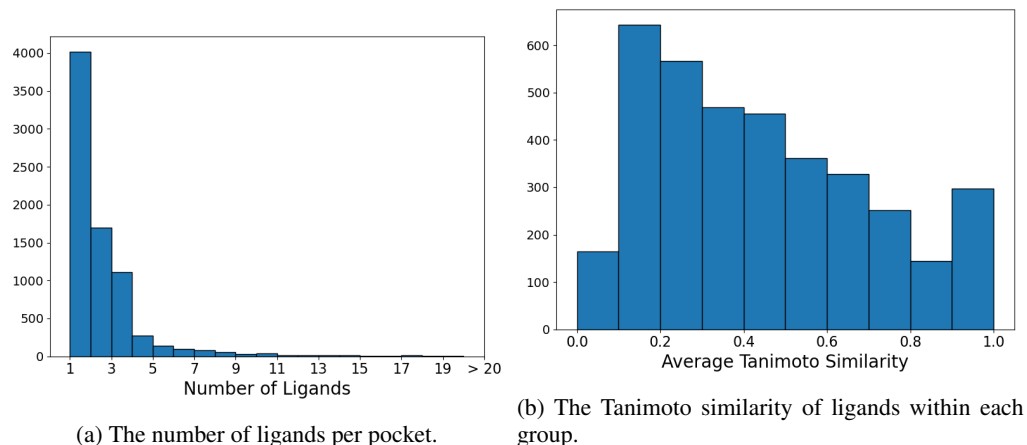

(a) The number of ligands per pocket.        (b) The Tanimoto similarity of ligands within each group.

Figure 6: Group Ligands Statistics in the PDBBind dataset.

---

**Algorithm 1** Procedure of preprocessing data for GROUPBIND

---

**Input:** A (unaligned) dataset with protein-ligand complexes $\{(\mathcal{P}^{(i)}, \mathcal{L}^{(i)})\}_{i=1}^N$.

**Output:** An aligned dataset $\{(\mathcal{P}^{(i)}, \mathcal{L}_{1:G_i}^{(i)})\}_{i=1}^M$. Each datapoint has an aligned protein $\mathcal{P}^{(i)}$ and a group of ligands $\mathcal{L}_{1:G_i}^{(i)}$ that can bind to this protein.

 1: Group complexes initially by their proteins' UniProt IDs and names, which are provided in the raw PDBBind dataset.
 2: Further cluster proteins by aligning their amino acid sequences and then performing agglomerative clustering Ward Jr (1963) with the optimal number of clusters, which is determined using silhouette scores Tibshirani et al. (2001).
 3: Align proteins in each group by finding the longest common subsequence of amino acids and apply the Kabsch algorithm Kabsch (1976) to obtain the optimal rototranslation. Compute pairwise alignment RMSDs.
 4: Based on the alignment RMSD, select the protein with minimum average RMSD to others as the reference protein. For other proteins, apply to complexes with the solved optimal rigid transformations relative to the reference protein.
 5: Cluster grouped complexes further by ligand centers to make sure group ligands are in the same pocket. Repeat alignment steps 3 and 4.
 6: Isolate complexes as single groups that have too close ($< 0.4$ Å) or too far ($> 3.0$ Å) minimum protein-ligand distances.

---

## C  EXPERIMENTAL DETAILS

### C.1  MODEL DETAILS

We use the same way to construct the protein-ligand heterogeneous graph as DIFFDOCK and follow the same node and edge featurization to make sure the performance improvement come from our introduced GROUPBIND paradigm. The triangle attention module between protein and group ligands is employed at the beginning of each layer, with 4 attention heads and 128-dimension key / query / value hidden features. The model consists of 5 layers with 48 scalar features and 10 vector features, except that in the ablation study Section 4.4, we use a smaller model with 24 scalar features and 6 vector features.

### C.2  TRAINING DETAILS

Since our GROUPBIND operates on the pocket level, we set a smaller sigma 10.0 for the translational noise. The other diffusion related parameters remain same as DIFFDOCK. For the auxiliary distance loss (Section 7), we map the distance to 40 bins from 2 Å to 22 Å, and the distance loss is weighted by 0.1 and added to the score matching loss. We train our model with AdamW Loshchilov & Hutter (2017) optimizer with the learning rate of 0.001 and the weight decay of 10.0.

## D  ADDITIONAL RESULTS

Following prior work, we also evaluate the top-1 and top-5 centroid distance in Table 4. It can be seen that our models are consistently better than all previous baselines. Augmentation with ligands from the training set also improves the model's performance in terms of centroid distances. We also provide the full results of ablation studies (Section 4.4) and different numbers of augmented ligands (Section 4.3) in Table 5 and Table 6 for further reference.

## E  TRAINING AND INFERENCE TIME

We trained our model on eight NVIDIA A100 GPUs for 350 epochs. Each epoch takes about 9min so the total training time is about 52.5 hrs. In comparison, DiffDock is trained on four 48GB RTX A6000 GPUs for 850 epochs (around 18 days). Considering the performance of A100 is about twice that of A6000, the actual training cost for our model and DiffDock is approximately 1:2. The

Table 4: Summary of top-1 and top-5 ligand centroid distance metrics of baseline models and GROUPBIND variants. (↑) / (↓) denotes a larger / smaller number is better. The best results are highlighted with **bold text**.

| Method | Top-1 Centroid Distance | | | | | Top-5 Centroid Distance | | | | |
| | Percentiles (↓) | | | % < thres. (↑) | | Percentiles (↓) | | | % < thres. (↑) | |
| | 25th | 50th | 75th | 5Å | 2Å | 25th | 50th | 75th | 5Å | 2Å |
|---|---|---|---|---|---|---|---|---|---|---|
| GNINA | 0.8 | 3.7 | 23.1 | 53.6 | 40.2 | 0.6 | 2.0 | 8.2 | 66.8 | 49.7 |
| SMINA | 1.0 | 2.6 | 16.1 | 59.8 | 41.6 | 0.6 | 1.9 | 6.2 | 72.9 | 50.8 |
| TANKBIND | 0.9 | 1.8 | 4.4 | 77.1 | 55.1 | 0.8 | 1.4 | 2.9 | 86.8 | 62.0 |
| P2RANK+SMINA | 0.8 | 2.6 | 14.8 | 60.1 | 44.1 | 0.6 | 1.8 | 12.3 | 66.2 | 53.4 |
| P2RANK+GNINA | 0.6 | 2.2 | 14.6 | 60.9 | 48.3 | 0.5 | 1.4 | 9.2 | 69.3 | 57.3 |
| EQUIBIND+SMINA | 0.7 | 2.1 | 7.3 | 69.3 | 49.2 | 0.5 | 1.3 | 5.1 | 74.9 | 58.9 |
| EQUIBIND+GNINA | 0.6 | 1.9 | 9.9 | 66.5 | 50.8 | 0.5 | 1.1 | 5.3 | 73.7 | 60.1 |
| DIFFDOCK | 0.6 | 1.4 | 3.2 | 80.6 | 60.4 | 0.4 | 0.7 | 1.8 | 89.8 | 78.7 |
| GROUPBIND-P2RANK-S | **0.5** | **1.0** | 3.1 | 81.2 | 67.6 | **0.3** | **0.6** | 1.3 | 91.1 | 82.3 |
| GROUPBIND-P2RANK-A | **0.5** | **1.0** | **2.4** | **83.7** | **71.5** | **0.3** | **0.6** | **1.2** | **93.1** | **84.5** |
| GROUPBIND-Ref-S | 0.4 | 0.8 | 1.6 | 94.5 | 80.3 | 0.3 | 0.6 | 1.1 | 96.7 | 88.1 |
| GROUPBIND-Ref-A | 0.4 | 0.8 | 1.5 | 94.7 | 84.5 | 0.3 | 0.6 | 1.0 | 97.2 | 90.6 |

Table 5: Full results of GROUPBIND ablation studies.

| Method | MP-GL | Att | Dist | Percentiles (↓) | | | % < thres. (↑) | |
| | | | | 25th | 50th | 75th | 5Å | 2Å |
|---|---|---|---|---|---|---|---|---|
| DIFFDOCK | | | | 1.76 | 2.98 | 5.21 | 73.3 | 30.6 |
| | ✓ | | | 1.70 | 2.51 | 4.33 | 83.9 | 35.2 |
| GROUPBIND | ✓ | ✓ | | 1.49 | 2.50 | 3.98 | 84.2 | 38.8 |
| | ✓ | ✓ | ✓ | 1.39 | 2.23 | 4.10 | 84.2 | 42.9 |

Table 6: Full results with different maximum numbers of ligands in each group.

| Method | # Ligs | Percentiles (↓) | | | % < thres. (↑) | |
| | | 25th | 50th | 75th | 5Å | 2Å |
|---|---|---|---|---|---|---|
| GROUPBIND-Ref-S | 1 | 1.09 | 1.86 | 3.32 | 87.5 | 53.5 |
| | 2 | 1.02 | 1.70 | 3.20 | 88.6 | 57.9 |
| | 3 | 1.04 | 1.74 | 3.19 | 87.8 | 55.4 |
| | 4 | 1.07 | 1.73 | 2.96 | 89.5 | 56.5 |
| | 5 | 1.03 | 1.72 | 3.16 | 88.9 | 56.5 |
| GROUPBIND-Ref-A | 2 | 1.00 | 1.66 | 3.18 | 88.1 | 56.8 |
| | 3 | 0.99 | 1.71 | 3.18 | 89.2 | 56.2 |
| | 4 | 1.04 | 1.6 | 3.07 | 90.0 | 59.6 |
| | 5 | 1.02 | 1.7 | 2.93 | 90.0 | 59.0 |

reduced training time for our model can be attributed to training at the pocket-level and grouping ligands, which leads to a smaller number of training data points compared to DiffDock.

The average time for our method to generate 40 samples per pocket is 30s (with a maximum number of ligands in a group set to 5). If we are interested in docking poses of multiple different ligands for the same protein, the actual average inference time for each ligand will be lower.

| Method | Inference Time (s) |
|---|---|
| GNINA | 127 |
| SMINA | 126 |
| TANKBIND | 0.7 / 2.5 |
| P2RANK + SMINA | 126 |
| P2RANK + GNINA | 127 |
| EQUIBIND + SMINA | 126 |
| EQUIBIND + GNINA | 127 |
| DiffDock | 40 |
| Ours (per pocket) | 30 |

Table 7: Inference time comparison between different docking methods.

# F  ADDITIONAL EXPERIMENTS

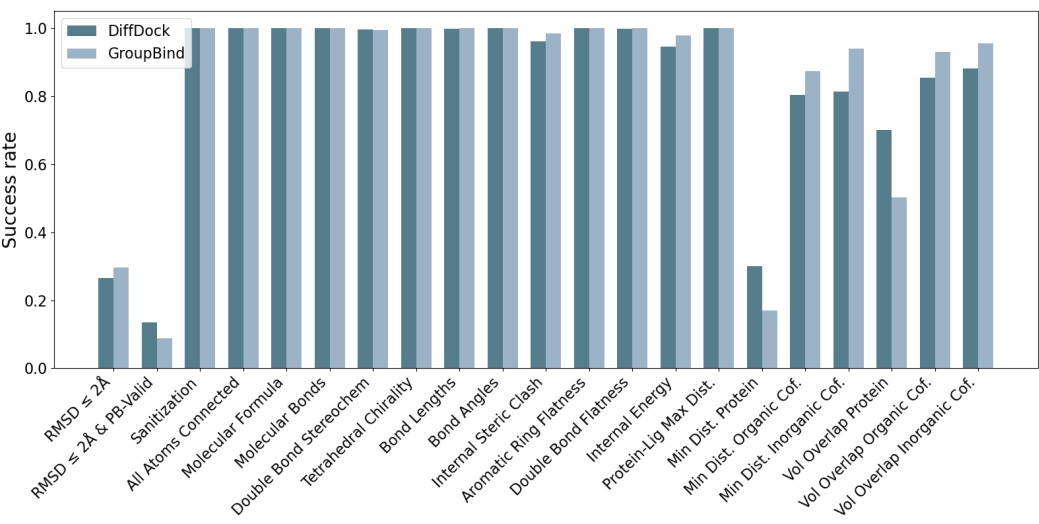

Figure 7: PoseBuster Results

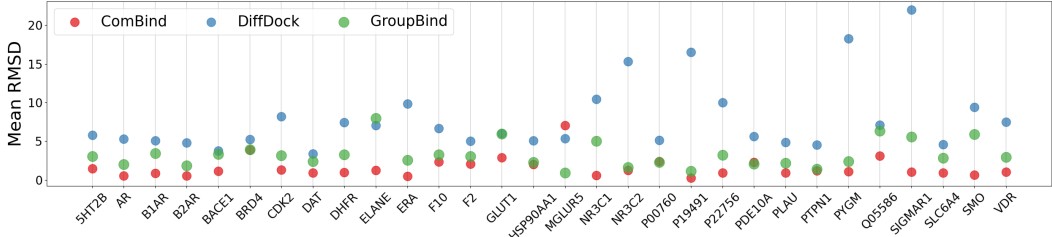

Figure 8: ComBind Results

We further compare our model and DiffDock under the PoseBuster benchmark (Buttenschoen et al., 2024) in Figure 7. Our model achieves a higher overall success rate than DiffDock, though with a lower PB-Valid success rate. Upon closer investigation, we found that our generated docking poses may contain more instances of unreasonable protein-ligand distances than those generated by DiffDock, which could be a side effect of incorporating augmented ligands.

We also included the experiment on the ComBind benchmark (Paggi et al., 2021) in Figure 8. As shown, our method consistently outperforms DiffDock across these 30 distinct proteins, benefiting from the insight that multiple ligands binding to the same protein pocket tend to exhibit similar docking poses. However, our model still shows performance gaps compared to ComBind. On one hand, this advantage arises because ComBind defines the docking grid using the center of mass (CoM) of the reference ligand, allowing the method to leverage this information. On the other hand, we attribute this to the more limited generalization ability of DiffDock compared to classic physics-based docking methods like GLIDE, which is used in ComBind. Combining our framework with a more robust backbone model that offers better generalization remains a promising direction for future work.

