# OpenReview forum: "Group Ligands Docking to Protein Pockets"
_ICLR.cc/2025/Conference — ICLR 2025 Poster_

### Official Review · Reviewer_fL2Q · 2024-11-02

**Soundness:** 3
**Presentation:** 3
**Contribution:** 3
**Rating:** 8
**Confidence:** 4

**Summary:**

The paper describes an approach for leveraging the insight that similar ligands that bind to the same protein target are expected to binding similarly.  A method for attending between ligands as part of a diffusion docking process is described and the results convincingly show the benefit of this approach.

**Strengths:**

Particularly during the lead optimization phase, it is reasonable to expect that there are a number of known ligands for the target protein and using this information to improve docking performance could help in the discovery of better ligands.

A reasonable approach for message passing across ligands with triangle attention is described.  This might provide a more general template for other tasks where output are represented as graphs and there is a known consistency bias.

This approach does not require the accessory ligands to have a known structure at inference time, which is a realistic scenario.

Informative ablation studies are performed.

**Weaknesses:**

The contributions are inappropriate as written.  This is not the first time the concept that multiple similar molecules can be used to enhance molecular docking has been described.  ComBind (Paggi et al, 2021; cited in the paper) does exactly that.  There are other methods that use this insight in different ways (e.g. selecting poses from ensemble docking).  The contributions should be qualified that this is the first end-to-end deep neural network approach to molecular docking that uses this insight.

The results only compare to single-ligand docking.  Comparing to ComBind (or OpenComBind if lacking a Schroedinger license) would be more relevant.  Can the diffusion model approach make better use of the similarity bias than previous methods? This question goes unanswered.

Not going beyond the PDBbind to identify alternative augmentation ligands (of which there are many, as structures aren't needed for inference) is a missed opportunity that weakens the paper as with the current evaluation framework many ligands can't be put into groups.  Replicating the ComBind evaluation would provide predefined groups of ligands while also making it possible directly compare to a conventional approach.

There are issues with using a time split to assess generalizability, but as this is the same split used with DiffDock it is appropriate to use for the comparisons performed here.

The overloading of "k" to mean two different things in Fig 2 is confusing.

I found equations (4) and (5) confusing due to the overloading of z - the text says these two separate values (presumably equivalent to AF2 incoming and outgoing triangle attention edges) will be stacked, while the equations say they are summed.

"and since the C-C bond length is about 1.5A."

Table 2 is apparently showing Top-10 results, but this fact is only stated in the main text. Why not replicate the reporting in Table 1?  "Med." isn't defined, but is presumably median RMSD.

4.4: "Section" is used when "Figure" is meant.  "ifself"

**Questions:**

How does your method compare to ComBind in a head-to-head comparison? Answering this would significantly increase my enthusiasm for the paper as, in addition to providing a more relevant baseline, it would involve testing using ligands without known structure as augmentation ligands (assuming the ComBind evaluation framework is used).

---

> ### Author Response · Authors · 2024-11-25
> **Response to Reviewer fL2Q**
>
> Thank you for your insightful and constructive comments as well as your appreciation of our work in **leveraging the insight that similar ligands that bind to the same protein target are expected to binding similarly**. Below are some clarifications and answers to your questions. If our response does not fully address your concerns, please post additional questions; we will be happy to discuss further.
>
> ## Q1: The contributions are inappropriate as written. This is not the first time the concept that multiple similar molecules can be used to enhance molecular docking has been described. ComBind (Paggi et al, 2021; cited in the paper) does exactly that. There are other methods that use this insight in different ways (e.g. selecting poses from ensemble docking). The contributions should be qualified that this is the first end-to-end deep neural network approach to molecular docking that uses this insight.
>
> Thank you for highlighting relevant work [1,2] that utilizes the concept of similar molecules to enhance docking. We have incorporated these references **into the Related Work** section for completeness. Additionally, we have clarified our contributions as follows: ```"This is the first time the concept that multiple molecules should yield similar docking poses has been integrated into an end-to-end deep neural network approach for the molecular docking task."```
>
>
>
> ## Q2: The results only compare to single-ligand docking. Comparing to ComBind (or OpenComBind if lacking a Schroedinger license) would be more relevant. Can the diffusion model approach make better use of the similarity bias than previous methods? This question goes unanswered. Replicating the ComBind evaluation would provide predefined groups of ligands while also making it possible directly compare to a conventional approach.
>
> Thank you for your valuable suggestion! We have included the experiment on the ComBind benchmark ***in Appendix F, Figure 8***, which includes 248 experimentally determined ligand-binding poses across 30 proteins representing all major families of drug targets. By replicating the the ComBind evaluation framework, as shown, our method consistently outperforms DiffDock across these 30 distinct proteins, benefiting from the insight that multiple ligands binding to the same protein pocket tend to exhibit similar docking poses. However, our model still shows performance gaps compared to ComBind. We attribute this to the more limited generalization ability of diffusion-based generative models (as discussed in [3,4]) compared to classic physics-based docking methods like GLIDE, which is used in ComBind. Combining our framework with a more robust backbone model that offers better generalization remains a promising direction for future work.
>
> ## Q3:There are issues with using a time split to assess generalizability, but as this is the same split used with DiffDock it is appropriate to use for the comparisons performed here.
>
> There are indeed challenges with using a time-based split to assess generalizability. However, since this split is consistent with the setup used in DiffDock, it is appropriate for the comparisons conducted in this work. By evaluating both our baseline methods and proposed methods under the same setting, we ensure a fair and reliable comparison.
>
> ## Q4: The overloading of "k" to mean two different things in Fig 2 is confusing.
>
> Thank you for pointing this out. In Figure 2, $i, j, k$ refer to atoms, $L_k$ denotes the k-th ligand, and $G_{L_k}$ represents the graph of the k-th ligand. We have carefully proofread the text to ensure that all notations are used consistently throughout the manuscript to avoid any confusion.
>
>
> ## Q5: I found equations (4) and (5) confusing due to the overloading of z - the text says these two separate values (presumably equivalent to AF2 incoming and outgoing triangle attention edges) will be stacked, while the equations say they are summed.
>
> They are stacked together -- the z has already been updated in the module defined by Eq(4), and then it serves as the input of the module defined by eq(5)
>
> ## Q6: Table 2 is apparently showing Top-10 results, but this fact is only stated in the main text. Why not replicate the reporting in Table 1? "Med." isn't defined, but is presumably median RMSD.
>
> Thank you for pointing this out. We have clarified in Table 2 that ```"The results of perfect selection under 10 samples are reported." ```Additionally, we have defined ```"Median."``` as Median RMSD and explicitly included the Median RMSD values in ***Table 2*** for consistency with Table 1.
>
> ## Q7: 4.4: "Section" is used when "Figure" is meant. "ifself"
>
> We have carefully reviewed and updated all references to figures, tables, and sections in the manuscript. "Figure" is now consistently used for images, and "Table" is used for tabular data.

---

> > ### Comment · Reviewer_fL2Q · 2024-11-25
> >
> > Is the worse performance relative to combind because you are doing blind docking while they are docking to the binding site?

---

> > > ### Author Response · Authors · 2024-11-26
> > > **Response to Reviewer fL2Q (Regarding Combind Result)**
> > >
> > > Thank you for your insightful comments on ComBind's superior results. In the ComBind protocol, they utilize a 15 Å inner box and a 30 Å outer box as the input for GLIDE docking, which can be considered a form of blind docking.
> > >
> > > Another likely reason for ComBind's better performance is the use of a larger number of helper ligands (approximately 20 per target), whereas GroupBind relies solely on self-group ligands.
> > >
> > > Also, as discussed in the PoseBuster Benchmark [1], traditional methods consistently outperform DL-based approaches, highlighting the need for significant improvements in future deep-learning docking methods.
> > >
> > > [1] Buttenschoen, Martin, Garrett M. Morris, and Charlotte M. Deane. "PoseBusters: AI-based docking methods fail to generate physically valid poses or generalise to novel sequences." Chemical Science 15.9 (2024): 3130-3139.

---

> > > > ### Comment · Reviewer_fL2Q · 2024-11-26
> > > >
> > > > It is a little worrisome that you think a 15Å box is equivalent to blind docking - it most certainly is not.  If the paper is accepted, please provide more information on your Combind comparison, since it would be reasonable to think you would use the same sets of ligands as Combind, but that apparently is not what you did.

---

> ### Author Response · Authors · 2024-11-25
> **Response to Reviewer fL2Q (References)**
>
> ## References:
>
> [1] Paggi, Joseph M., et al. "Leveraging nonstructural data to predict structures and affinities of protein–ligand complexes." Proceedings of the National Academy of Sciences 118.51 (2021): e2112621118.
>
> [2] McNutt, Andrew T., and David R. Koes. "Open-ComBind: harnessing unlabeled data for improved binding pose prediction." Biophysical Journal 123.3 (2024): 186a.
>
> [3] Corso, Gabriele, et al. "Deep confident steps to new pockets: Strategies for docking generalization." ArXiv (2024).
>
> [4] Buttenschoen, Martin, Garrett M. Morris, and Charlotte M. Deane. "PoseBusters: AI-based docking methods fail to generate physically valid poses or generalise to novel sequences." Chemical Science 15.9 (2024): 3130-3139.

---

> ### Author Response · Authors · 2024-11-27
> **Response to reviewer fL2Q about concerns on blind docking**
>
> Thank you for your thoughtful feedback and for pointing out an important oversight in our previous response. We apologize for any confusion caused by the omission of key details.
>
> To clarify, ComBind uses the center of mass (CoM) of the entire protein as the docking center. Our earlier interpretation of blind docking was based on the absence of prior binding site knowledge. Since ComBind uses the protein's CoM as the docking center, we initially assumed it does not rely on binding site information. However, upon further reflection, we recognize that the strict definition of blind docking includes an additional requirement: the docking region must encompass the entire protein to allow unbiased exploration of all potential binding sites. By this stricter definition, ComBind would only qualify as blind docking if the bounding box were large enough to cover the entire protein, rather than being restricted to a smaller region (e.g., 15–30 Å). We acknowledge this oversight and thank the reviewer for bringing it to our attention.
>
> It remains notable that ComBind achieves such strong results without prior binding site information. We agree that aligning our method's ligand and pocket selection criteria to ComBind would strengthen this aspect of our study. If the paper is accepted, we will revise the manuscript to include these details, ensuring greater transparency and addressing any potential confusion.
>
> We sincerely appreciate your constructive feedback and will incorporate the necessary revisions to enhance the clarity and rigor of our work.

---

> > ### Comment · Reviewer_fL2Q · 2024-11-27
> >
> > ComBind uses the center of the reference ligand to define the docking grid (https://github.com/drorlab/combind/blob/main/dock/grid.py).  If they were centering a 15/30 A^3 grid at the protein CoM the binding site wouldn't be in the box for many of the targets.

---

> > > ### Author Response · Authors · 2024-12-03
> > >
> > > Dear Reviewer fL2Q,
> > >
> > > Thank you very much for clarifying the docking grid definition in ComBind! We sincerely apologize for our earlier misunderstanding and the resulting miscommunication. Upon reviewing the source code you referenced and revisiting our analysis, we now recognize that ComBind defines the docking grid using the CoM of the reference ligand, not CoM of the protein as we previously stated. This correction makes the results of ComBind much more reasonable, as centering the grid at the protein's CoM would indeed exclude the binding site for many targets.
> > >
> > > The confusion on our end arose from a miscommunication among the authors regarding the meaning of CoM in this context. We appreciate your patience in pointing this out and allowing us the opportunity to correct this error.
> > >
> > > We will update the manuscript to accurately describe ComBind's method for defining the docking grid and will ensure this distinction is clear in the discussion of our experiments. We are also preparing to train a reference pocket based DiffDock and GroupBind to create a more equitable comparison with ComBind in a future version of the manuscript.
> > >
> > > Thank you again for bringing this to our attention and for your constructive feedback, which has been invaluable in improving the rigor and accuracy of our work.

---

### Official Review · Reviewer_J3p9 · 2024-11-03

**Soundness:** 3
**Presentation:** 3
**Contribution:** 2
**Rating:** 5
**Confidence:** 5

**Summary:**

This paper presents a deep learning model for molecular docking, which improves data utilization and model quality by docking multiple molecules to the same protein. It further enhances the connections between similar atoms across different molecules and the same protein amino acids using a triangular perceptual network. Through this multi-molecule docking approach, the model surpasses existing methods.

**Strengths:**

The study proposes a new molecu lar docking framework to simultaneously consider multiple ligands docking to a protein.

**Weaknesses:**

Overall, this paper offers a certain level of contribution, but the experimental section requires clearer descriptions and further discussion. For the figures in the paper, the authors should provide detailed captions, including explanations of the methods used.

**Questions:**

1.	The authors should account for the effect of protein similarity on the results by performing redundancy removal on the test set proteins that are either duplicated or highly similar to those in the training or validation sets. Tools like MMseqs or other alternatives could be used for this process.
2.	The authors need to explain how the similarity between ligands in the group ligand set impacts the results.
3.	How would the results of GROUPBIND change if tested on the latest PoseBuster (version 1 and version 2)?
4.	In Figure 4 on page 8, does the success rate refer to Top 1 or Top 40? What do “SG” and “AG” specifically mean? These details should be clarified in the figure caption.
5.	What does Figure 5 on page 8 illustrate, and where is it referenced in the article?
6.	In Figure 6 on page 8, does the success rate refer to Top 1 or Top 40? The meanings of “NG”, “SG”, and “AG” should also be clarified in the figure caption.
7.	On line 457 on page 9, the percentage 36.3% is mentioned twice, which could cause confusion and should be clarified.

---

> ### Author Response · Authors · 2024-11-25
> **Reponse to Reviewer J3p9**
>
> Thank you for your insightful and constructive comments as well as your appreciation of our work in **improving data utilization and model quality by docking multiple molecules to the same protein**. Below are some clarifications and answers to your questions. If our response does not fully address your concerns, please post additional questions; we will be happy to discuss further.
>
> ## Q1: The authors should account for the effect of protein similarity on the results by performing redundancy removal on the test set proteins that are either duplicated or highly similar to those in the training or validation sets. Tools like MMseqs or other alternatives could be used for this process.
>
> Following DiffDock's benchmark, we employed a time-based split for the training, validation, and testing sets. This approach inherently ensures redundancy removal by preventing test set proteins that are duplicated or highly similar to those in the training or validation sets from being included.
>
> Additionally, in ***Appendix Secion F (Figure 8)***, we present results from new experiments conducted on the ComBind benchmark. In this setup, proteins are clustered to ensure minimal similarity between groups [1], resulting in 248 experimentally determined ligand-binding poses across 30 proteins representing all major families of drug targets. We compare the performance of DiffDock and GroupBind on these 30 proteins with minimal sequence or structural similarity. Our results show that GroupBind consistently outperforms DiffDock under these conditions, demonstrating the robustness of our model.
>
>
> ## Q2: The authors need to explain how the similarity between ligands in the group ligand set impacts the results.
>
> As shown in ***Figure 4(c)***, knowledge of dissimilar ligands that can dock to the same pocket provides informative signals to the model, generally resulting in lower docking RMSD (Spearman correlation = -0.39). This suggests that grouping dissimilar ligands benefits our model by enhancing its ability to identify correct protein-ligand interactions, ultimately leading to more accurate docking poses. Additionally, we have included a case in ***Appendix A*** to further illustrate the motivation behind our method.
>
> ## Q3: How would the results of GROUPBIND change if tested on the latest PoseBuster?
>
> ***In Appendix G, Figure 7***, we present a comparison of our model with DiffDock under the PoseBuster benchmark [2]. Our model achieves a higher overall success rate than DiffDock, although it shows a lower PB-Valid success rate. Upon closer examination, we observe that our generated docking poses may contain more unreasonable protein-ligand distances compared to those produced by DiffDock. This could be a side effect of incorporating augmented ligands into our approach.
>
>
> ## Q4: In Figure 4 on page 8, does the success rate refer to Top 1 or Top 40? What do “SG” and “AG” specifically mean? These details should be clarified in the figure caption. In Figure 6 on page 8, does the success rate refer to Top 1 or Top 40? The meanings of “NG”, “SG”, and “AG” should also be clarified in the figure caption.
>
> In Figure 4 (b)(c)(d), the success rate corresponds to the Top 40 ligand poses. We have clarified this in the caption by stating that ```"the best docking pose among 40 candidates is used to calculate the success rate."``` Additionally, we have explicitly defined the abbreviations in the caption: ```"w/SG, w/AG, and w/o G indicate the subsets where augmented ligands from the test set, training set, and no augmented ligands are used, respectively. SG, AG, and NG represent the corresponding model performance under these different ligand augmentation settings."```
>
>
> ## Q5: What does Figure 4 on page 8 illustrate, and where is it referenced in the article?
>
> Figure 3 (we reordered figures in the new version) illustrates an example of docking poses generated by GroupBind and DiffDock, demonstrating that ```"reference ligands tend to bind the target protein with similar poses, which is effectively captured by GroupBind, resulting in lower average RMSD compared to DiffDock."``` This figure is referenced in Section 4.2 as: ```"We also provide examples of docking poses generated by GroupBind compared to DiffDock in Figure 4."```
>
> ## Q6: On line 457 on page 9, the percentage 36.3% is mentioned twice, which could cause confusion and should be clarified.
>
> We apologize for the minor typographical error. It has been corrected to: ```"When provided with the native reference pocket, the top-1 ligand RMSD is significantly improved to 36.6% and 36.3%, respectively." ```

---

> > ### Author Response · Authors · 2024-11-25
> > **Reponse to Reviewer J3p9 (References)**
> >
> > ## References:
> >
> > [1] Paggi, Joseph M., et al. "Leveraging nonstructural data to predict structures and affinities of protein–ligand complexes." Proceedings of the National Academy of Sciences 118.51 (2021): e2112621118.
> >
> > [2] Buttenschoen, Martin, Garrett M. Morris, and Charlotte M. Deane. "PoseBusters: AI-based docking methods fail to generate physically valid poses or generalise to novel sequences." Chemical Science 15.9 (2024): 3130-3139.

---

### Official Review · Reviewer_rEyK · 2024-11-04

**Soundness:** 3
**Presentation:** 3
**Contribution:** 3
**Rating:** 6
**Confidence:** 5

**Summary:**

This paper introduces GroupBind, a blind rigid docking method predicated on the biochemical observation that ligands binding to the same target protein often adopt similar poses. GroupBind employs an interaction layer for a group of ligands and a triangle attention module to embed protein-ligand and group-ligand pairs. Performance is evaluated on the PDBbind dataset.

**Strengths:**

1. The idea of leveraging similar binding poses among ligands targeting the same protein is intriguing and biologically relevant.
2. The experimental results suggest that incorporating augmented ligands improves docking performance.

**Weaknesses:**

1. The core idea is similar  to MCS (Maximum Common Substructure) docking [1, 2], which assume that ligands with similar substructures exhibit similar docking poses. GroupBind, however, assumes all ligands share similar docking poses. Figure 1 depicts highly similar ligands with similar docking structures. Conversely, [3] (Figure 4) illustrates cases where structurally distinct ligands adopt distinct poses. A statistical comparison quantifying the difference between the MCS docking assumption and GroupBind's assumption is warranted.

2. Comparing GroupBind-Ref against blind docking methods like DiffDock is unfair. DiffDock performs blind docking, whereas GroupBind-Ref utilizes prior knowledge of the binding pocket, making it a site-specific docking method. This introduces a significant advantage for GroupBind-Ref.

3. The evaluation should include more baselines such as FABind [5] and FABind+ [6], for which source code is available. Expanding the evaluation to include datasets like PoseBuster would assess GroupBind's ability to predict physically plausible structures.

4. The reported top-1 docking success rate for DiffDock (32.4% with 40 samples in Table 1) appears considerably lower than previously
reported results (38.2% in [4] and 36.0% in [5]). This discrepancy requires clarification.

5. The clarity of the writing could be improved. Specific points of confusion are detailed below in the Questions section.

[1] A High-Quality Data Set of Protein-Ligand Binding Interactions Via Comparative Complex Structure Mod, 2024

[2] FitDock: protein-ligand docking by template fitting, 2022

[3] DeltaDock: A Unified Framework for Accurate, Efficient, and Physically Reliable Molecular Docking, 2024

[4] DiffDock: Diffusion Steps, Twists, and Turns for Molecular Docking, 2022

[5] FABind: Fast and Accurate Protein-Ligand Binding, 2023

[6] FABind+: Enhancing Molecular Docking through Improved Pocket Prediction and Pose Generation, 2024

**Questions:**

1.	The definition of "ligand ground graph" needs explicit formalization. The current lack of clarity makes understanding this crucial concept challenging. For example, when referring “noisy group ligands” in line 214, it is difficult to understand this concept.
2.	The references to tables and figures are confusing. Avoid using "Section" to denote figures, tables, and section of manuscript at the same time. Use "Figure" when referring to images (e.g., Figure 1, Figure 2) and "Table" for tabular data (e.g., Table 1, Table 2).
3.	A thorough proofread is necessary to correct spelling and grammatical errors. For instance, "beyound" on line 101 should be corrected to "beyond." A comprehensive review of the entire text is recommended.
4.	Figure 7's inclusion of Figures 3, 4, 5, and 6. For better visual organization and clarity, these figures should be presented as subfigures within a single figure (e.g., Figure 7a, 7b, 7c, and 7d). This allows for easier comparison and a more streamlined presentation.
5.	Lacking explanation of "NG" in Figure 6. While the meaning of "NG" in Figure 6 might be discernible from the main text, its meaning should be explicitly stated. This ensures clarity and avoids requiring readers to search through the text for an explanation. A concise definition or clarification of "NG" within the caption is crucial.

---

> ### Author Response · Authors · 2024-11-25
> **Reponse to Reviewer rEyK (Part 1)**
>
> Thank you for your insightful and constructive comments as well as your appreciation of our work that **leveraging similar binding poses among ligands targeting the same protein is intriguing and biologically relevant**. Below are some clarifications and answers to your questions. If our response does not fully address your concerns, please post additional questions; we will be happy to discuss further.
>
> ## Q1: MCS docking assumes that ligands with similar substructures exhibit similar docking poses, while GroupBind assumes all ligands share similar docking poses. Conversely, Figure 3 illustrates cases where structurally distinct ligands adopt distinct poses. A statistical comparison quantifying the difference between the MCS docking assumption and GroupBind's assumption is warranted.
>
> This is a very good question! To clarify the motivation behind our method, we have added an example in ***Appendix A Section A***. Specifically, PDB entries 6g2c and 6g29 contain distinct ligands (Tanimoto similarity = 0.26) that bind to the same pocket. Despite their differences, several key interactions are conserved: for instance, a hydrogen bond is formed between N3 in 6g2c / N12 in 6g29 and an oxygen atom in the side chain of a pocket residue, while π-stacking interactions occur between the ring C7C8N9C10N11 in 6g2c / ring C8N7N6C4C3C2 in 6g29 and two benzene rings in the side chains of pocket residues. These conserved interactions ultimately result in similar binding poses for these distinct ligands.
>
> ***In Appendix B, Figure 6(b)***, we present the average Tanimoto similarity of ligands within each group, revealing that while some pockets do bind similar ligands, a larger number accommodate dissimilar ones. And as demonstrated in ***Figure 4(c)***, grouping dissimilar ligands helps our model better capture relevant protein-ligand interactions, leading to more accurate docking poses. **In Section 4.5**, we further highlights the improvement provided by our model even when augmented ligands are unavailable.
>
> **Overall, our approach relies on a more general assumption than the MCS docking assumption [1,2], and our model—leveraging the capabilities of deep neural networks—effectively handles these more diverse scenarios**. DeltaDock's Figure 4 [5] illustrates cases where structurally distinct ligands adopt distinct poses, but the context is how to handle "large" binding pockets, which is a valuable finding and doesn't contradict with our assumption. Studying how to better handle large binding pockets in our framework would be an interesting future direction.
>
> ## Q2: Comparing GroupBind-Ref against blind docking methods like DiffDock is unfair.
>
> We did not compare GroupBind-Ref with DiffDock, as GroupBind-Ref utilizes additional reference pocket information. The bold text annotations in Table 1 indicate the best results achieved in the blind docking setting. We have updated the table caption to clarify this. **GroupBind-Ref variants are not taken into consideration for a fair comparison**.
>
> ## Q3: The evaluation should include more baselines such as FABind and FABind+ , for which source code is available. Expanding the evaluation to include datasets like PoseBuster would assess GroupBind's ability to predict physically plausible structures.
>
> Thank you for highlighting FABind [3] and FABind+ [4]! **We appreciate the suggestion and have now included these works in the Related Work section to acknowledge their contributions**. However, since our model is based on a diffusion-based generative docking framework, we chose not to include FABind and FABind+ in the main results table to maintain a fair comparison. Integrating our framework with other cutting-edge docking methods is an exciting direction for future work. Given that our model demonstrates significant improvements when combined with diffusion-based docking approaches, we believe our results showcase both the novelty and promise of our method.
>
> Additionally, we have compared our model with DiffDock on the PoseBustesr benchmark [6] in ***Appendix Section F***. Our model achieves a higher overall success rate than DiffDock, though with a lower PB-Valid success rate. Upon closer investigation, we found that our generated docking poses may contain more instances of unreasonable protein-ligand distances than those generated by DiffDock, which could be a side effect of incorporating augmented ligands.

---

> ### Author Response · Authors · 2024-11-25
> **Reponse to Reviewer rEyK (Part 2)**
>
> ## Q4: The reported top-1 docking success rate for DiffDock (32.4% with 40 samples in Table 1) appears considerably lower than previously reported results (38.2% in [4] and 36.0% in [5]). This discrepancy requires clarification.
>
> We strictly adhered to the sampling procedure outlined in the DiffDock repository to generate the results presented in our manuscript. To facilitate further examination, we have provided the sampling results and a simple notebook for quick evaluation in this anonymous link: https://drive.google.com/file/d/1awPMogbZpXOBIEo1YpzWWLkZpP3lRIgI/view. One possible cause of discrepancies may stem from RDKit initialization issues, which could impact the accuracy of local structures. However, since both our model and DiffDock were evaluated under the same machine setup, we believe our comparison is fair and reliable.
> We are actively investigating the root cause of the performance discrepancy and will update both the DiffDock results and our own in Table 1 once we have further insights.
>
>
> ## Q5: The clarity of the writing could be improved. For instance, "beyound" on line 101 should be corrected to "beyond." A comprehensive review of the entire text is recommended.
>
> We have conducted a thorough proofread of the manuscript to address spelling and grammatical errors, including correcting "beyound" to "beyond". All sections have been carefully reviewed to improve overall language quality. We hope this revised version is clearer and easier to follow.
>
> ## Q6: The definition of "ligand ground graph" needs explicit formalization. The current lack of clarity makes understanding this crucial concept challenging. For example, when referring “noisy group ligands” in line 214, it is difficult to understand this concept.
>
> We have clarified the term ```"ligand ground graph"``` to explicitly describe its meaning: ```"We construct a graph across ligands in the group by placing all group ligands in the same coordinate system and adding edges between different ligands."```
>
> We have replaced the term ```"Noisy group ligands"``` with ```"diffused group ligands"``` to more accurately reflect their role in the training process. Specifically, "diffused group ligands" refers to ```"the ligands after noise is added during the diffusion process."```
>
> ## Q7: The references to tables and figures are confusing. Avoid using "Section" to denote figures, tables, and section of manuscript at the same time. Use "Figure" when referring to images (e.g., Figure 1, Figure 2) and "Table" for tabular data (e.g., Table 1, Table 2).
>
> We have carefully reviewed and updated all references to figures, tables, and sections in the manuscript. "Figure" is now consistently used for images, and "Table" is used for tabular data. This ensures clarity and avoids confusion for the reader.
>
> ## Q8: Figure 3's inclusion of Figures 3, 4, 5, and 6. For better visual organization and clarity, these figures should be presented as subfigures within a single figure (e.g., Figure 3a, 3b, 3c, and 3d).
>
> We have renumbered Figure 4 including 4a,4b,4c and 4d (as we reordered the figures in the revised version for better visual flow) to ensure consistent labeling throughout the manuscript. The numbering of subsequent figures has also been updated accordingly.
>
> ## Q9: Lacking explanation of "NG" in Figure 3d. While the meaning of "NG" in Figure 6 might be discernible from the main text, its meaning should be explicitly stated. This ensures clarity and avoids requiring readers to search through the text for an explanation. A concise definition or clarification of "NG" within the caption is crucial.
>
> We have added a new caption to clarify the meaning of "NG" directly in Figures 4d and 6. Specifically, ```"w/SG, w/AG, and w/o G" indicate subsets where augmented ligands from the test set, training set, and no augmented ligands are used```, respectively. Similarly, ```"SG, AG, and NG" represent the corresponding model performance under these different augmentation settings: test set, training set and no augmented ligands```. This addition ensures clarity and eliminates the need for readers to search the main text for definitions.

---

> > ### Author Response · Authors · 2024-11-25
> > **Response to Reviewer rEyK (References)**
> >
> > ## References:
> >
> > [1] Li X, Shen C, Zhu H, et al. A high-quality data set of protein–ligand binding interactions via comparative complex structure modeling[J]. Journal of Chemical Information and Modeling, 2024, 64(7): 2454-2466.
> >
> > [2] Yang X, Liu Y, Gan J, et al. FitDock: protein–ligand docking by template fitting[J]. Briefings in bioinformatics, 2022, 23(3): bbac087.
> >
> > [3] Pei Q, Gao K, Wu L, et al. Fabind: Fast and accurate protein-ligand binding[J]. Advances in Neural Information Processing Systems, 2024, 36.
> >
> > [4] Gao K, Pei Q, Zhu J, et al. FABind+: Enhancing Molecular Docking through Improved Pocket Prediction and Pose Generation[J]. arXiv preprint arXiv:2403.20261, 2024.
> >
> > [5] Yan, Jiaxian, et al. "Deltadock: A unified framework for accurate, efficient, and physically reliable molecular docking." arXiv preprint arXiv:2410.11224 (2024).
> >
> > [6] Buttenschoen, Martin, Garrett M. Morris, and Charlotte M. Deane. "PoseBusters: AI-based docking methods fail to generate physically valid poses or generalise to novel sequences." Chemical Science 15.9 (2024): 3130-3139.

---

> > > ### Comment · Reviewer_rEyK · 2024-11-25
> > >
> > > I thank the authors for their thorough and comprehensive rebuttal, which has effectively addressed all of my previous concerns. The revised manuscript, strengthened by the inclusion of compelling experimental results, now presents a convincing argument for the utility and significance of GroupBind. The authors' responsiveness to reviewer comments is commendable, and the overall contribution of GroupBind to the field is noteworthy. Future investigations exploring group docking in more complex scenarios, particularly those incorporating protein flexibility, represent a promising direction for continued research. Accordingly, I am pleased to raise my score.

---

> > > > ### Author Response · Authors · 2024-11-25
> > > > **Follow Up Response to Reviewer rEyK**
> > > >
> > > > Thank you for your positive and timely response—we deeply appreciate your willingness to reconsider your score. We believe there are several promising future directions worth exploring. For instance, incorporating protein pocket information and flexibility could further enhance docking accuracy. Additionally, as demonstrated in ComBind [1], the large collection of unstructured helper ligands available in the ChEMBL dataset [2] could also be leveraged to improve docking performance. Overall, we are confident that our framework can have a positive impact on the community and foster further advancements in this area.
> > > >
> > > > References:
> > > >
> > > > [1] Paggi, Joseph M., et al. "Leveraging nonstructural data to predict structures and affinities of protein–ligand complexes." Proceedings of the National Academy of Sciences 118.51 (2021): e2112621118.
> > > >
> > > > [2] Gaulton, Anna, et al. "ChEMBL: a large-scale bioactivity database for drug discovery." Nucleic acids research 40.D1 (2012): D1100-D1107.

---

### Official Review · Reviewer_Pwff · 2024-11-04

**Soundness:** 4
**Presentation:** 4
**Contribution:** 3
**Rating:** 8
**Confidence:** 3

**Summary:**

This paper presents a molecular docking framework called GROUPBIND, which enhances the binding capability of a ligand to a target protein pocket by leveraging other ligands that bind to the same pocket.
The framework introduces message padding among groups of ligands and a triangle attention module for protein-ligand pairs. Experimental results validate that GROUPBIND improves docking performance based on diffusion models.

**Strengths:**

(1) The writing and organization of this paper are very clear.

(2) This paper is intriguing because it is based on the idea of enhancing the binding capability of the current ligand by considering the binding positions of other ligands that target the same protein.

**Weaknesses:**

(1) Why are the results of DIFFDOCK in Table 1 worse than those in the original paper, and it seems that the bold text annotations might be inaccurate?

(2) From Figure 1, we can see that molecules binding to the same pocket indeed have similar structures, but how many pockets in the dataset exhibit this situation? Is there any statistical data on the number of pockets and the corresponding similar ligands in the PDBBind dataset?

(3) During inference, when searching the database for ligands similar to the query ligand, how many entries in the test set can retrieve similar ligands? If similar ligands cannot be retrieved, does that mean the model becomes ineffective?

**Questions:**

Refer to the content in the Weaknesses.

---

> ### Author Response · Authors · 2024-11-25
> **Response to Reviewer Pwff**
>
> Thank you for your insightful and constructive comments as well as your appreciation of our work in **leveraging other ligands that bind to the same pocket to enhance the binding prediction**. Below are some clarifications and answers to your questions. If our response does not fully address your concerns, please post additional questions; we will be happy to discuss further.
>
> ## Q1: Why are the results of DIFFDOCK in Table 1 worse than those in the original paper, and it seems that the bold text annotations might be inaccurate?
>
> We strictly adhered to the sampling procedure outlined in the DiffDock repository (https://github.com/gcorso/DiffDock) to generate the results presented in our manuscript. To facilitate further examination, we have provided the sampling results and a simple notebook for quick evaluation in this anonymous link: https://drive.google.com/file/d/1awPMogbZpXOBIEo1YpzWWLkZpP3lRIgI/view. One possible cause of discrepancies may stem from **RDKit initialization issues**, which could impact the accuracy of local structures. However, since both our model and DiffDock were evaluated **under the same machine setup**, we believe our comparison is fair and reliable. We are actively investigating the root cause of the performance discrepancy and will update both the DiffDock results and our own in Table 1 once we have further insights.
>
> Regarding the bold text annotations, we refrained from comparing GroupBind-Ref to other baselines to maintain a fair comparison, as it leverages additional reference pocket information. The bold text in Table 1 highlights the best-performing model **within the blind docking setting**. We have updated the caption of Table 1 to clarify this distinction.
>
> ## Q2: From Figure 1, we can see that molecules binding to the same pocket indeed have similar structures, but how many pockets in the dataset exhibit this situation? Is there any statistical data on the number of pockets and the corresponding similar ligands in the PDBBind dataset?
>
> In Figure 6(a), we present a histogram showing the number of ligands per pocket. There are 7697 unique pockets for the 19k complexes in the PDBBind dataset. Among these, 3,681 pockets (47.8%) contain more than one ligand. Additionally, Figure 6(b) illustrates the Tanimoto similarity of ligands within each group (with two or more ligands). This analysis shows that while some pockets are indeed bonded with similar ligands, a larger portion of pockets contain ligands that are dissimilar. Despite this, our model effectively handles such cases. Specifically, as shown in Figure 4(c), the presence of dissimilar ligands binding to the same pocket provides valuable information to our model, leading to improved docking accuracy and generally lower RMSD values (Spearman correlation = -0.39). This suggests that **grouping dissimilar ligands is advantageous for our model in discerning correct interactions with proteins**, ultimately yielding more accurate docking poses.
>
> For further illustration, we have added a case study in **Appendix A Section A** to clarify the motivation behind our method. Specifically, PDB entries 6g2c and 6g29 contain distinct ligands (Tanimoto similarity = 0.26) that bind to the same pocket. Despite their differences, several key interactions are conserved: for instance, a hydrogen bond is formed between N3 in 6g2c / N12 in 6g29 and an oxygen atom in the side chain of a pocket residue, while π-stacking interactions occur between the ring C7C8N9C10N11 in 6g2c / ring C8N7N6C4C3C2 in 6g29 and two benzene rings in the side chains of pocket residues. These conserved interactions ultimately result in similar binding poses for these distinct ligands.
>
> ## Q3: During inference, when searching the database for ligands similar to the query ligand, how many entries in the test set can retrieve similar ligands? If similar ligands cannot be retrieved, does that mean the model becomes ineffective?
>
> As described in Section 4.1 (Experimental Setup), the 363 test ligands are organized into 86 groups containing more than one ligand (255 ligands in total) and 108 groups with a single ligand. By expanding the augmented ligand database to include the training set, we can construct additional ligand groups for 32 out of the 108 single-ligand groups.
>
> In cases where binding information is unavailable for a target protein, an alternative strategy is to use ligands that bind to homologous proteins, as discussed in Section 4.5, where we also observe performance improvement. If neither ligands binding to the same protein nor those binding to homologous proteins are accessible, our approach defaults to the backbone model (DiffDock in this study). This fallback ensures that the model remains applicable and that performance is minimally impacted.

---

### Author Response · Authors · 2024-11-25
**Global Response to All Reviewers**

We sincerely thank all reviewers for their valuable and constructive feedback, which has been instrumental in improving our work. Below, we summarize the key suggestions and our corresponding responses, as well as the updates made to the draft:
1. **Results of DiffDock baselines and our sampled results (Pwff, rEyK)**:
We strictly followed the sampling procedure outlined in the DiffDock repository (https://github.com/gcorso/DiffDock) to generate the results presented in our manuscript. To ensure reproducibility, we have uploaded the sampling results and a simple notebook for quick evaluation in this anonymous link: https://drive.google.com/file/d/1awPMogbZpXOBIEo1YpzWWLkZpP3lRIgI/view. One potential cause of discrepancies could be related to RDKit initialization issues, which may affect the accuracy of local structures. However, since both our model and DiffDock were evaluated under the same machine setup, we believe our comparison is fair and reliable.

2. **Motivation and statistical results of our methods (Pwff, rEyK)**:
To better illustrate the motivation behind our method, we have added an example in ***Appendix Section A***. Additionally, we have included detailed statistical analyses and figures of our group data in ***Appendix Section B***.

3. **Relevant related work (rEyK, fL2Q)**:
We have incorporated recent related works, such as FABind, FABind+, and physics-based methods like ComBind, which use similar docking ligands to enhance docking accuracy. These works have been discussed in the ***Related Work*** section.

4. **Results on the PoseBuster and ComBind benchmarks (rEyK, J3p9, fL2Q)**:
We have extended our evaluations to include comparisons with DiffDock and ComBind on the PoseBuster and ComBind benchmarks. These results are presented in **Appendix Section F**.

5. **Clarifications on demonstrations and captions (rEyK, J3p9, fL2Q)**:
To enhance understanding, we have included more detailed and precise explanations for our demonstrations and improved the clarity of figure captions throughout the manuscript.

---

### Meta-Review · Area_Chair_HHeg · 2024-12-18

**Metareview:**

In this submission, the authors propose a new ligand-protein blind docking method. The problem is important for drug discovery and development, and the proposed method is interesting. In particular, grouping ligands for docking is reasonable and inspiring for the community, which may trigger more new methodologies in the future.

At the same time, although the authors made great efforts to resolve the reviewers' concerns, some reviewers still have questions about 1) the rationality of time-based data splitting and 2) the rationality of the criterion for defining "blinding docking." Especially for the time-based data splitting, Reviewer J3p9 and AC think that this strategy cannot remove the redundancy between the training and testing sets in terms of protein similarity. The authors should discuss the rationality and/or the potential risk of this strategy in the final version of the paper.

Overall, this submission has advantages that outweigh its weaknesses, but the authors should revise it carefully based on their comments.

**Additional Comments On Reviewer Discussion:**

Three of the four reviewers interacted with the authors. In the discussion phase, AC discussed with Reviewer J3p9 about the acceptance of the submission.

---

### Decision · Program_Chairs · 2025-01-22

Accept (Poster)